# Ncx3-Induced Mitochondrial Dysfunction in Midbrain Leads to Neuroinflammation in Striatum of A53t-α-Synuclein Transgenic Old Mice

**DOI:** 10.3390/ijms22158177

**Published:** 2021-07-30

**Authors:** Rossana Di Martino, Maria Josè Sisalli, Rossana Sirabella, Salvatore Della Notte, Domenica Borzacchiello, Antonio Feliciello, Lucio Annunziato, Antonella Scorziello

**Affiliations:** 1Department of Neuroscience, Division of Pharmacology, Reproductive and Dentistry Sciences, School of Medicine, Federico II University of Naples, Via Pansini 5, 80131 Naples, Italy; rossanadim@gmail.com (R.D.M.); mariajose.sisalli@unina.it (M.J.S.); sirabell@unina.it (R.S.); sa.dellanotte@gmail.com (S.D.N.); 2Department of Molecular Medicine and Medical Biotechnology, Federico II University of Naples, Via Pansini 5, 80131 Naples, Italy; domenica.borzacchiello@unina.it (D.B.); feliciel@unina.it (A.F.); 3IRCCS SDN, Via E. Gianturco 113, 80143 Naples, Italy; lannunzi@unina.it

**Keywords:** Parkinson’s disease, mitochondrial dysfunction, neuroinflammation, α-synuclein

## Abstract

The exact mechanism underlying selective dopaminergic neurodegeneration is not completely understood. The complex interplay among toxic alpha-synuclein aggregates, oxidative stress, altered intracellular Ca^2+^-homeostasis, mitochondrial dysfunction and disruption of mitochondrial integrity is considered among the pathogenic mechanisms leading to dopaminergic neuronal loss. We herein investigated the molecular mechanisms leading to mitochondrial dysfunction and its relationship with activation of the neuroinflammatory process occurring in Parkinson’s disease. To address these issues, experiments were performed in vitro and in vivo in mice carrying the human mutation of α-synuclein A53T under the prion murine promoter. In these models, the expression and activity of NCX isoforms, a family of important transporters regulating ionic homeostasis in mammalian cells working in a bidirectional way, were evaluated in neurons and glial cells. Mitochondrial function was monitored with confocal microscopy and fluorescent dyes to measure mitochondrial calcium content and mitochondrial membrane potential. Parallel experiments were performed in 4 and 16-month-old A53T-α-synuclein Tg mice to correlate the functional data obtained in vitro with mitochondrial dysfunction and neuroinflammation through biochemical analysis. The results obtained demonstrated: 1. in A53T mice mitochondrial dysfunction occurs early in midbrain and later in striatum; 2. mitochondrial dysfunction occurring in the midbrain is mediated by the impairment of NCX3 protein expression in neurons and astrocytes; 3. mitochondrial dysfunction occurring early in midbrain triggers neuroinflammation later into the striatum, thus contributing to PD progression during mice aging.

## 1. Introduction

Parkinson’s disease (PD) is a chronically progressive, age-related neurodegenerative pathology characterized by resting tremor, rigidity, bradykinesia, gait disturbance, postural instability and dementia [1]. A major neuro-pathological feature of the disease is a degeneration of dopaminergic neurons in the substantia nigra pars compacta (SNpc) and in other brainstem regions [2,3]. Lewy bodies are the second neuropathological feature of PD. These are eosinophilic cellular inclusions comprising a dense core of filamentous material, which mainly consists of α-synuclein [4,5,6].

Several mechanisms have been advocated as pathogenetic in PD. They may take place inside the degenerating neurons and are identified as cell-autonomous processes, or they can occur outside the degenerating neurons and are identified as non-cell-autonomous processes. In the last decade, the alterations in mitochondrial function have been claimed among the most common cell-autonomous mechanisms leading to neuronal degeneration [7,8,9,10]. The electrophysiological activity of the nigral dopaminergic neurons plays a fundamental role in generating these processes [11,12,13]. Indeed, dopaminergic neurons are characterized by a pacemaking activity responsible for generating action potentials also in the absence of synaptic input. This activity exposes neurons to large Ca^2+^ transients [13,14] with consequent overstimulation of several classes of ionic channels and transporters, including L-type voltage-dependent-and cyclic nucleotide-sensitive channels and the plasma membrane transporters including the sodium calcium exchangers (NCXs), in order to counteract the alteration of intracellular Ca^2+^ homeostasis and to prevent dopaminergic neuronal demise [15]. The recent observation that cells deficient in complex-I showed an alteration in cytosolic calcium handling, reduced mitochondrial calcium accumulation and ATP synthesis [12,16] led us to hypothesize a possible relationship between mitochondrial dysfunction and perturbation of intracellular calcium homeostasis as a pathogenetic factor involved in PD progression. In this regard, it is worth noting that plasmalemmal NCX2 and NCX3, two proteins playing a key role in the regulation of cytosolic calcium concentrations ([Ca^2+^]_c_) in physiological and pathological conditions [17,18,19,20,21,22], might contribute to mitochondrial Na^+^/Ca^2+^ exchange in human dopaminergic neurons, thus preventing neurodegeneration caused by mitochondrial Ca^2+^ (_m_Ca^2+^) overload [23]. More interestingly, in primary mesencephalic neurons from A53T transgenic mice embryos, the downregulation of NCX3 levels is linked to mitochondrial depolarization and _m_Ca^2+^ increase compared to wild type neurons, further supporting the hypothesis that mitochondrial dysfunction in PD is linked to _m_Ca^2+^ mishandling [22,24]. On the other hand, among the non-cell-autonomous processes, the interaction between neuronal and non-neuronal cells is considered the main additional pathogenetic mechanism involved in neurodegeneration occurring in PD. In fact, PD post-mortem brains also display increased gliosis [25,26], both in terms of a rise in the number of glial cells and in terms of the activity of astrocytes and microglial cells [27,28,29,30,31]. However, the molecular mechanisms responsible for microglial activation as well as its role in the pathogenesis of neurodegeneration occurring in PD progression are still matters of debate, since post-mortem studies do not allow us to clearly assess whether neuroinflammation is a cause or consequence of neuronal degeneration [32]. Our group provided evidence that the isoform 1 of the sodium calcium exchanger NCX (NCX1) plays a key role in microglial activation in ischemic rat brain [33] as well as in striatum of A53T TG mice, an animal model of familial form of PD [22]. In this regard, we observed that the number of IBA-1 and GFAP-positive cells was increased in the striatum of 12-month-old A53T TG mice compared to WT mice, and that the IBA1-positive cells overexpressed NCX1. Interestingly, in the midbrain of these mice the number of glial fibrillary acidic protein (GFAP)-positive cells increased compared to WT mice without any changes in the number of IBA-1-positive cells and in the expression of NCX1. Nevertheless, in this brain region the number of dopaminergic neurons decreased as well as the expression of NCX3 [22]. Considering these premises and considering that Ca^2+^ signaling is relevant to promote glial cells’ activation [33,34,35] as well as neurodegeneration [20,36,37,38], we explored the molecular intracellular events which correlate mitochondrial dysfunction in glial and neuronal cells with the progression of PD. In particular, we investigated the possible relation between the expression and activity of NCX3 in the different cellular populations with mitochondrial dysfunction and neuroinflammatory response activation, as a pathogenetic mechanism leading to dopaminergic neuronal degeneration in mice expressing the human A53T variant of α-synuclein (A53T-α-syn) during aging. This mutation makes the protein more prone to aggregate [39] and causes severe motor deficits leading to paralysis and death [40]. These animals also develop age-dependent α-syn inclusions that recapitulate the pathology seen in human PD patients [41,42]. Moreover, the aggregates interact efficiently with the plasma membrane, and in particular with cardiolipin, the principal component of mitochondrial membrane, in comparison to WT α-synuclein [43] and more interestingly, α-synuclein mutant forms, including A53T, stimulate proinflammatory cytokines’ production and microglia activation [44].

Therefore, the expression and activity of NCX isoforms were explored in neurons and glial cells obtained from A53T-α-syn mice. In these cells, mitochondrial function was monitored by confocal microscopy and fluorescent dyes in order to assess _m_Ca^2+^ content and mitochondrial membrane potential. Parallel experiments were also performed in 4 and 16-month-old A53T-α-syn mice in order to demonstrate the relevance of mitochondrial dysfunction and oxidative stress in neuroinflammatory response detected in A53T-α-syn mouse brain during aging.

## 2. Results

### 2.1. Mitochondrial Dysfunction in Neuronal Cells Obtained from A53t-α-Syn Mice Depends on Decrease in Ncx3 Expression

Experiments performed in mesencephalic neurons obtained from A53T-α-syn mouse embryos (E15) revealed mitochondrial membrane depolarization compared to WT neurons (Figure 1A). This finding is in line with data previously reported showing a decrease in NCX3 without any change in NCX1 protein expression accompanied by increases in cytosolic calcium concentration ([Ca^2+^]_i_) and mitochondrial calcium concentration ([Ca^2+^]_m_) [22]. Parallel experiments performed in mesencephalic astrocytes obtained from A53T-α-syn mice demonstrated a reduction in NCX3 expression compared to WT cells (Figure 1B). Interestingly, confocal microscopy performed in these cells revealed an increase in cytosolic calcium content associated with a rise in mitochondrial calcium concentration without any alteration in mitochondrial membrane potential (Figure 1C). Conversely, both in striatal neurons and astrocytes derived from A53T-α-syn no changes in mitochondrial function as well as in NCX1 and NCX3 expression were detected compared to WT cells (Appendix A). These findings led us to hypothesize that the reduction in NCX3 protein expression might be associated with an impairment of its activity, which consequently might lead to mitochondrial depolarization in neurons and not in astrocytes obtained from A53T-α-syn midbrain, thus suggesting NCX3-induced mitochondrial depolarization as a potential molecular mechanism leading to dopaminergic neuronal demise observed in PD.

### 2.2. Differences in Ncx3 Expression in Nigrostriatal Pathway Are Associated with Mitochondrial Impairment in A53t-α-Syn Transgenic Mice during Aging

Further experiments were performed ex vivo in midbrain and striatum obtained from 4 and 16-month-old A53T-α-syn and WT mice to demonstrate the mechanistic link between NCX3 protein expression and mitochondrial dysfunction as a potential pathogenetic mechanism leading to PD progression. As reported in Figure 2, a decrease in the expression of NCX3 was detected in the midbrain of 4-month-old A53T-α-syn (Figure 2B), whereas an increase in the expression of NCX1 was detectable in the striatum during mice aging (Figure 2C). To confirm that the impairment in NCX3 protein expression was responsible for mitochondrial dysfunction, the expression of cytochrome c (cyt c), a marker of mitochondrial damage, and of neuronal nitric oxide synthases (nNOS), a marker of mitochondrial oxidative stress, was measured in 4 and 16-month-old A53T-α-syn and WT mice (Figure 3). The results of these experiments revealed an increase in cyt c expression in the midbrain of A53T-α-syn mice which occurred already in the early stage of disease (4-month-old mice) (Figure 3A), while in the striatum of A53T-α-syn mice, the increase in cyt c was detectable only in 16-month-old mice (Figure 3C). Interestingly, in the midbrain of A53T-α-syn mice the increase in cyt c expression was accompanied by an increase in nNOS expression level (Figure 3B), whereas in the striatum of A53T-α-syn mice, an increase in nNOS protein expression occurred only in 16-month-old A53T-α-syn mice (Figure 3D). These results are in line with those obtained in vitro, suggesting a possible relationship between the level of NCX3 expression and mitochondrial dysfunction.

### 2.3. Mitochondrial Dysfunction Triggers Neuroinflammation in A53t-α-Syn Transgenic Mice during Aging

To support the hypothesis that mitochondrial dysfunction observed in mesencephalic and striatal neurons might trigger neuroinflammation during the late PD progression, further experiments were performed in 16-month-old A53T-α-syn and WT mice with the aim to evaluate the expression levels of pro-inflammatory proteins, such as the inducible nitric oxide synthases (iNOS) and the Interleukin 1 beta (IL-1β). Moreover, Western blot experiments were also performed to evaluate GFAP and IBA-1 protein expression in 16-month-old A53T-α-syn and WT mice. The results obtained confirmed the increase of GFAP in A53T-α-syn mice both in the midbrain and in the striatum (Figure 4A,E), whereas IBA-1 increased only in the striatum of A53T-α-syn mice (Figure 4B,F). Interestingly, in this brain area, an increase in the expression of iNOS and IL-1β was observed in A53T-α-syn, whereas no changes in inflammatory protein expression were detected in the midbrain of A53T-α-syn adult mice compared to WT (Figure 4C–H).

## 3. Discussion

The results of the present study demonstrate that mitochondrial dysfunction occurring in mesencephalic neurons obtained from A53T-α-syn mice is associated with dopaminergic neuronal demise and consequently to a late activation of neuroinflammation in the nigrostriatal pathway. Moreover, these results also confirm the potential role of NCX3 in triggering mitochondrial dysfunction and neuroinflammatory response in a PD animal model. Indeed, in the striatum obtained from WT and A53T-α-syn mice no differences in NCX3 protein expression are detected during aging, and mitochondrial function is preserved. This finding is in line with data previously reported [20], showing that NCX3, apart its localization at plasma membrane level, is also localized on the outer mitochondrial membrane and promotes mitochondrial calcium efflux in physiological and pathological conditions, thus playing a pivotal role in the maintenance of intracellular Na^+^ and Ca^2+^ homeostasis not only in brain ischemia but also in neurodegenerative diseases [17,36,37,45,46,47,48,49]. More interesting, the finding that NCX3 is differently expressed in dopaminergic neurons in the midbrain compared to striatum led us to speculate a new potential mechanism contributing to the selective neuronal degeneration observed in PD. This result allows us to exclude the hypothesis that the increasing load of aggregated pathological form of α-synuclein, described in this model [39,40,42] might have a detrimental role through the regulation of NCX3 activity and expression. Indeed, due to their pacemaking function, mesencephalic neurons are more frequently exposed to continued calcium transients that make them more prone to mCa^2+^ overload and, consequently, to mitochondrial dysfunction [13,50,51,52,53]. In this scenario, the impairment of the expression and activity of NCX3 in mesencephalic neurons might contribute to their selective vulnerability in the midbrain of A53T-α-syn mice, due to NCX3′s contribution to the perturbation of [Ca^2+^]_i_ concentration, as already hypothesized by Sirabella et al. (2018) [22]. Therefore, the experiments performed in the present study further confirm that NCX3-dependent mitochondrial dysfunction occurring in mesencephalic neurons from A53T-α-syn might be responsible for the neuronal damage with consequent late activation of the cellular events along the nigrostriatal pathway that, in turn, might be responsible for PD progression. Indeed, Western blot experiments performed in 4 and 16-month-old A53T-α-syn and WT mice showed an increase in Cyt c and nNOS protein expression, both markers of mitochondrial damage and oxidative stress, respectively, that was already detectable in the midbrain of 4-month-old A53T-α-syn mice and was still elevated in the late stage of disease compared to WT mice. Conversely, in the striatum of A53T-α-syn mice, in which NCX3 expression did not change, in comparison to WT mice, an increase in Cyt c and nNOS protein levels occurred only in the late stage of the disease. This is in line with previously described immunohistochemistry experiments performed in 12-month-old A53T-α-syn mice showing a reduction in TH expression in the midbrain of these mice compared to WT [22]. Therefore, the results of the present study suggest that a progressive impairment of the oxidative metabolism occurring in the early stage in mesencephalic neurons might be claimed as a pathogenetic mechanism leading to dopaminergic neuronal damage responsible for the late progression of PD. Indeed, no increase in pro-inflammatory proteins was detected in 4-month-old A53T-α-syn both in the midbrain and in the striatum (data not shown), although mitochondrial dysfunction occurred. This finding supports the data recently reported in the literature that correlate mitochondrial dysfunction to the triggering of the neuroinflammatory process in PD. Indeed, damaged mitochondria can release numerous pro-inflammatory factors which, triggering microglial activation, lead to neuroinflammation in the nigrostriatal pathway [54,55,56]. This hypothesis is confirmed by the results presented in this study regarding the increase in IBA-1, IL-1β and iNOS detected in the striatum of 16-month-old A53T-α-syn mice, in which an increase in Cyt c and nNOS occurs. Moreover, the activation of microglial cells in the striatum of aged A53T-α-syn mice is associated with an increase in GFAP, thus confirming a relationship between mitochondrial-induced neuroinflammation and glial activation [57,58,59,60]. Conversely, in the midbrain of 16-month-old A53T-α-syn mice the increase in Cyt c and nNOS protein expression is accompanied by glial proliferation, as confirmed by the increase in GFAP protein levels without microglial activation. These findings, in accordance with data previously reported in 12-month-old A53T-α-syn mice [22], further support the hypothesis that in the midbrain, the activation of glial cells might be a consequence of the cellular responses triggered into the striatum of transgenic mice, and in turn, it might further contribute to dopaminergic neuronal demise observed in the midbrain during PD progression. Moreover, the data obtained in the striatum also suggest that the molecular mechanisms involved in the activation of microglial cells are different from those occurring in astrocytes. In fact, it is possible to hypothesize that the neuronal damage related to the impairment in mitochondrial membrane permeability in the midbrain since the early stages of the disease might promote the release of neuronal toxic factors able to stimulate the activation of microglial cells in the striatum of A53T-α-syn adult mice. Once activated, microglial cells can release pro-inflammatory cytokines in adult mice and in turn, promote astroglia proliferation in the striatum and in the midbrain, as confirmed by the increase in NCX1 protein expression described in these two brain areas in aged A53T-α-syn mice (Figure 5). Interestingly, in astrocytes obtained from midbrain of A53T-α-syn mice mitochondria, although they contain higher calcium compared to WT cells, they are not depolarized, thus suggesting their ability to support cellular proliferation. These findings let us hypothesize that the normal expression of NCX1 in transgenic astrocytes by maintaining [Ca^2+^]_i_ within physiological range is able to preserve mitochondrial function and, consequently, to promote gliosis in adult mice, despite the impairment of NCX3 protein expression. In this scenario, NCX1 overexpression would not be directly related to a detrimental effect in glial cells. This is in line with data previously reported in the ischemic brain [33] and might explain the consequent increase in dopaminergic neuronal injury observed in the midbrain of A53T-α-syn aged mice (Figure 5). On the other hand, the increased NCX1 expression occurring in striatum of A53T-α-syn mice being a reflection of glial proliferation might represent a useful druggable target to modulate neuroinflammation in PD.

In conclusion, the results reported in the present study demonstrate that mitochondrial dysfunction in dopaminergic neuronal cells might exert a detrimental role in PD progression. Indeed, mitochondrial dysfunction occurring in mesencephalic neurons in A53T-α-syn mice at the early stage of the disease promotes neuronal degeneration and activates microglial cells in the striatum. The activated microglia can in turn promote pro-inflammatory factors’ release in the striatum of these mice with consequent glial activation and progressive impairment of dopaminergic neuronal plasticity in the late stage of the disease.

## 4. Materials and Methods: In Vivo and In Vitro Models

### 4.1. A53t-α-Syn Transgenic Mice

Mice that express human A53T-α-synuclein under the control of prion promoter (PrP-SNCA*A53T) [40] were obtained from The Jackson Laboratory. Mice hemizygous for the A53T mutation were bred on a mixed C57Bl/6 × C3H background to produce transgenic and non-transgenic littermates. In all cases, 4 and 16-month-old transgenic mice were directly compared with age-matched wild type littermates. To identify transgenic mice PCR amplifications were performed according to the protocol provided by The Jackson Laboratory. Mice were group-housed (1–5 animals/cage) in temperature and humidity-controlled rooms under a 12-h light–dark cycle and fed an ad libitum diet of standard mouse chow. Experiments were performed on male and female mice according to the international guidelines for animal research and approved by the Animal Care Committee of “Federico II” University of Naples, Italy.

### 4.2. Primary Neurons from A53t-α-Syn Mice

Primary midbrain and striatal cultures were isolated from 15-day-old A53T-α-syn and WT mouse brain embryos and prepared by modifying the previously described method of Fath and collaborators [61]. The mesencephalic and striatal tissues were minced and incubated with a dissection medium containing EBSS, DNAse, BSA and ovomucoid for 30 min at 37 °C. After incubation, the suspension from the two brain regions was centrifuged and subjected to mechanical dissection in order to obtain a cellular suspension. Then, the cells were placed on Poly-d-lysine-coated (100 µg/mL) plastic dishes, in MEM/F12 culture medium containing glucose, 5% deactivated fetal bovine serum, 5% horse serum, glutamine (2 mM), penicillin (50 U/mL), and streptomycin (50 μg/mL). The day after plating, cells were treated with Cytosine-β-d-arabinose-furanoside in vitro (10 μM) to prevent the non-neuronal cell growth. Neurons were cultured at 37 °C in a humidified 5% CO_2_ atmosphere and used after 10 days in vitro (DIV) for all experiments described. For confocal experiments, cells were plated on glass coverslips coated with Poly-d-lysine [62].

### 4.3. Primary Astrocytes from A53t-α-Syn Mice

Primary astrocyte cultures from midbrain and striatal A53T-α-syn and WT mice were obtained from the midbrain and striatum of newborn pups 1–2 days old (P1–2). The respective areas of the brain were aseptically dissected, and meninges were carefully removed. The tissues were minced finely and enzymatically (0.25% trypsin and DNAse/MgSO_4_ 37 °C, 30 min) and mechanically dissociated to produce single-cell suspensions. Then, the cell suspension was centrifuged at 1000 r.p.m. for 5 min and suspended in DMEM containing 100 U/mL penicillin, 100 μg/mL streptomycin and 10% FBS. The cells were plated in tissue flasks, then cultured at 37 °C in a 95% air 5% CO_2_ incubator. The medium change occurred once every two days. When cells grew to confluence (7–8 days), the flask was shaken with PBS three times to remove the loosely attached contaminated microglia. The attached enriched astrocytes were subsequently detached using trypsin-EDTA and then subjected to different experimental procedures.

For Western blotting experiments astrocytes were cultured in a T75 flask at 37 °C in a humidified 5% CO_2_ atmosphere and used after 14–15 days in culture (DIV). For confocal microscopy experiments, mature cells at 14–15 (DIV) were plated in glasses pre-coated with Poly-d-Lys and analyzed after 24 h.

### 4.4. Western Blot Analysis

Mouse brain tissue and primary neuronal cells were lysed in a buffer containing: Tris-HCl (20 mM, pH 7.5); NaF 10 mM; NaCl 150 mM; phenylmethylsulphonyl fluoride (PMSF) 1 mM; NONIDET P-40 1%; Na3VO4 1 mM; aprotinin 0.1%; pepstatin 0.7 mg/mL e leupeptin 1 μg/mL. Homogenates were centrifuged at 14,000 rpm for 20 min at 4 °C. The supernatant was used to perform Western blot analysis. Protein levels were determined using the Bradford method. The total protein amount used for each sample was 50 μg and it was separated on 8% or 15% sodium dodecyl sulfate-polyacrylamide gels with 5% sodium dodecyl sulphate stacking gel (SDS-PAGE) and electrotransferred onto Hybond ECL nitrocellulose paper (Amersham, Milan, Italy). The membranes were blocked in 5% non-fat dry milk in 0.1% Tween 20 (TBS-T; 2 mmol/L Tris HCl, 50 mmol/L NaCl, pH 7.5) for 1 h at room temperature (RT) and subsequently incubated overnight at 4 °C in the blocked buffer with the 1:1000 antibody for NCX1 (polyclonal rabbit antibody, Swant, Marly, Switzerland), 1:5000 antibody for NCX3 (polyclonal rabbit antibody, Philipson’s Laboratory, UCLA, Los Angeles, CA, USA), 1:1000 GFAP (polyclonal, Novus Biologicals, Littleton, CO, USA; NB300-141), 1:1000 for IBA-1 (polyclonal, Wako 019-19741), 1:1000 for nNOS (monoclonal, Santa Cruz, Dallas, TX, USA; NOS1(R-20)sc-648), 1:1000 for iNOS (monoclonal, NOS2, Santa CRUZ (C-11):sc-7271), 1:1000 for IL-1β (monoclonal, Cell Signaling, Danvers, MA, USA; 12242), 1:1000 for Cyt c (polyclonal, Cell Signaling 4272). Next, all membranes were washed 3 times with a solution containing Tween 20 (0.1%) and subsequently incubated with the secondary antibodies for 1 h (1:2000) at room temperature. Immunoreactive bands were detected by ECL (Amersham). Discrimination among the distinct types of extracts was ensured by running parallel Western blots with the endogenous β-actin or α-tubulin protein. The optical density of the bands was determined by the Image J program.

### 4.5. Confocal Microscopy and Mitochondrial Function

To assess the [Ca^2+^]_m_, neurons and astrocytes obtained from A53T-αsyn and WT mouse embryos and pups, respectively, were loaded with X-Rhod-1 (0.2 μM) for 15 min in a medium containing: 156 mM NaCl, 3 mM KCl, 2 mM MgSO_4_, 1.25 mM KH_2_PO_4_, 2 mM CaCl_2_, 10 mM glucose, and 10 mM HEPES. The pH was adjusted to 7.35 with NaOH. At the end of the incubation, cells were washed 3 times in the same medium. An increase in the mitochondria-localized intensity of fluorescence was indicative of mCa^2+^ overload [45].

[Ca^2+^]_c_ was measured using the fluorescent dye Fluo-3AM acetoxymethyl ester (Fluo-3AM). Cells were loaded with Fluo-3AM (5 nM) for 30 min at room temperature in the same medium described above. At the end of incubation, cells were washed 3 times in the same medium. An increase in [Ca^2+^]_c_ intensity of fluorescence was indicative of cytosolic Ca^2+^ overload [18]. The advantage to use fluo3 was that this calcium indicator can be loaded into the cells together with the mitochondrial calcium indicator X-Rhod-1, thus allowing a simultaneous comparison of calcium levels in the cytoplasmic and mitochondrial compartments. Mitochondrial membrane potential was assessed using the fluorescent dye tetramethylrhodamine ethyl ester (TMRE) in the “redistribution mode” [63]. Cells were loaded with TMRE (20 nM) for 30 min in the above-described medium. At the end of the incubation, cells were washed in the same medium containing TMRE (20 nM) and allowed to equilibrate. A decline in the mitochondria-localized intensity of fluorescence was indicative of mitochondrial membrane depolarization [63].

Confocal images were obtained using Zeiss inverted 700 confocal laser scanning microscopy and a 63× oil immersion objective. The illumination intensity of 543 Xenon laser used to excite X-Rhod-1 and TMRE, and of 488 Argon laser used to excite Fluo-3AM fluorescence, was kept to a minimum of 0.5% of laser output to avoid phototoxicity.

### 4.6. Statistical Analysis

Data were generated from a minimum of 3 independent experimental sessions for in vitro studies. Calcium and mitochondrial membrane potential measurements were performed at least in 200 cells for each of the 3 independent experimental sessions. Data were expressed as mean percentage ± S.E.M. Statistical comparisons between transgenic and their respective controls during aging were performed using the one-way ANOVA test followed by Newman–Keuls test. The unpaired Student’s T-test was used to analyse the trend inside the same group corresponding at the single specific age. *p*-value < 0.05 was considered statistically significant.

## 5. Conclusions

In conclusion, the results reported in the present study demonstrate that mitochondrial dysfunction described in mesencephalic neurons of A53T-α-syn mice at the early stage of the disease promotes neuronal degeneration and activates neuroinflammation in the striatum of these mice with consequent glial activation and progressive impairment of dopaminergic neuronal plasticity in the late stage of the disease.

## Figures and Tables

**Figure 1 ijms-22-08177-f001:**
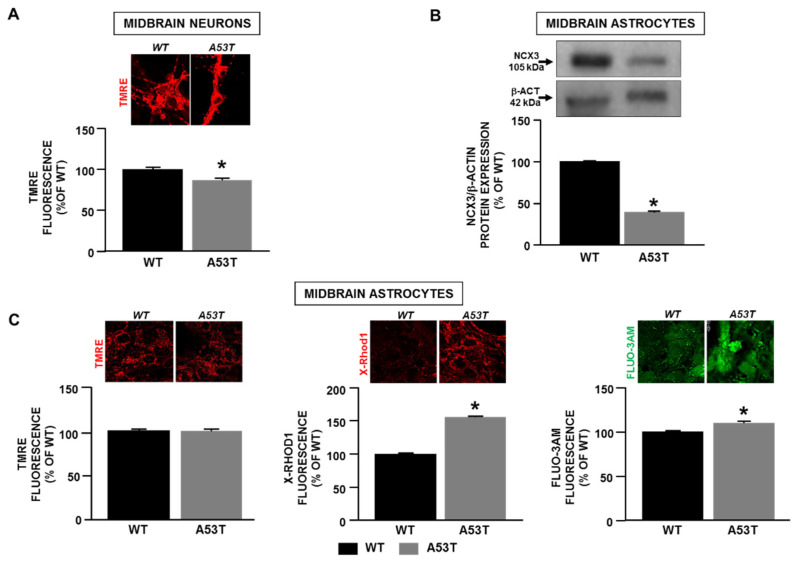
Mitochondrial function and NCX3 expression in mesencephalic neuronal cells from A53T-α-syn and WT mouse embryos/pups. (**A**) Quantification of the changes in mitochondrial membrane potential measured in primary mesencephalic neurons from A53T-α-syn and WT mouse embryos by confocal microscopy. (**B**) Quantification of NCX3 expression in primary mesencephalic astrocytes from A53T-α-syn and WT mouse pups (P1). (**C**) Quantification of mitochondrial membrane potential, [Ca^2+^]_m_ and [Ca^2+^]_c_ in primary astrocytes from A53T-α-syn and WT mouse pups by confocal microscopy (P1). In **A** and **C**: each bar represents the mean ± S.E.M. of the percentage of fluorescence intensity values of at least 20–30 neurons recorded in three independent experimental sessions. In **B**: each bar represents the mean ± S.E.M of the percentage of the NCX3 protein expression obtained in three independent experimental sessions. * *p* < 0.05 vs. WT neurons/astrocytes.

**Figure 2 ijms-22-08177-f002:**
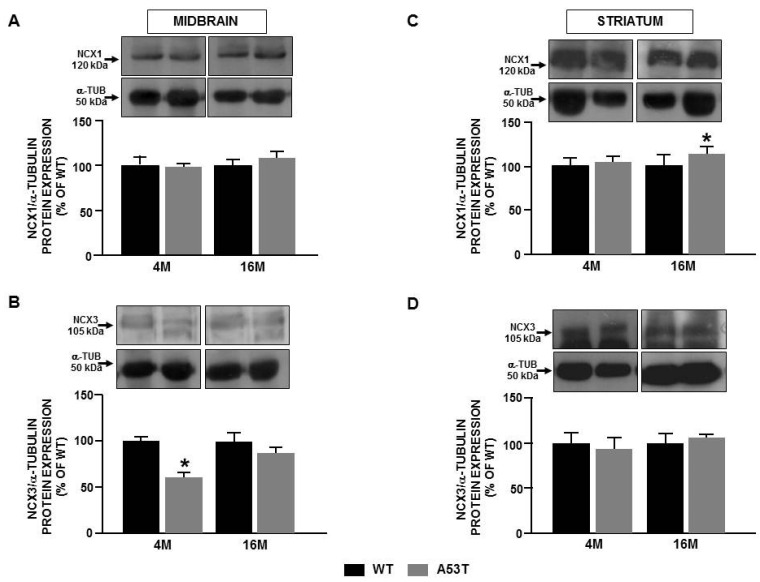
NCX1 and NCX3 expression in midbrain and striatum obtained from 4 and 16-month-old A53T-α-syn and WT mice. Western blot analysis of NCX1 and NCX3 protein expression in midbrain (**A**,**B**) and striatum (**C**,**D**) from A53T-α-syn and WT mice. Each bar represents the mean ± S.E.M. of the percentage of different experimental values obtained in three independent experimental sessions. * *p* < 0.05 compared to WT mice at the same age.

**Figure 3 ijms-22-08177-f003:**
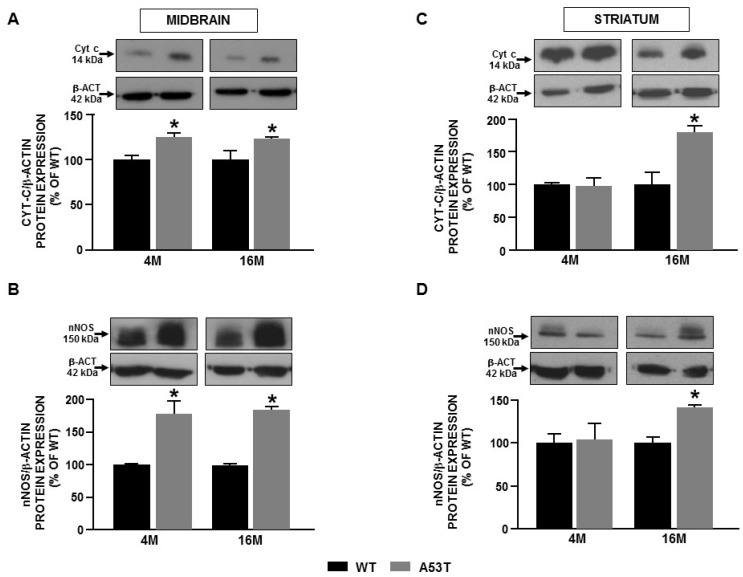
Cyt c and nNOS expression in midbrain and striatum obtained from 4 and 16-month-old A53T-α-syn and WT mice. Western blot analysis of Cyt c and nNOS protein expression in midbrain (**A**,**B**) and striatum (**C**,**D**) from A53T-α-syn and WT mice. Each bar represents the mean ± S.E.M. of the percentage of different experimental values obtained in three independent experimental sessions. * *p* < 0.05 compared to WT mice at the same age.

**Figure 4 ijms-22-08177-f004:**
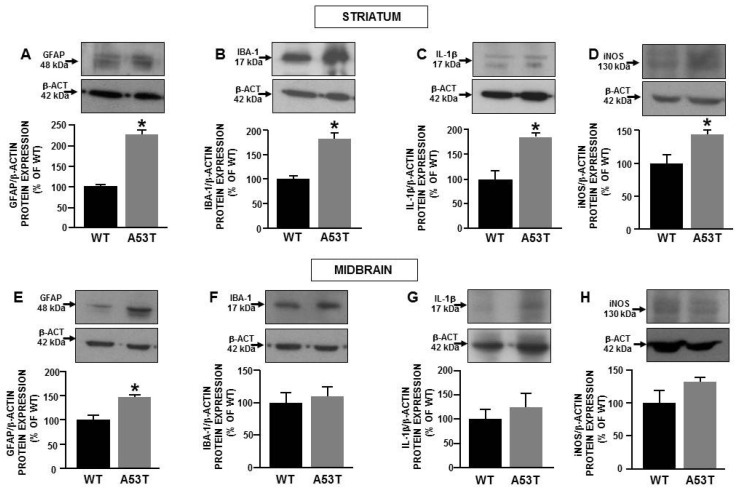
Activation of neuronflammation in 16-month-old A53T-α-syn mice. Western blot analysis of GFAP, IBA-1, iNOS, and IL-1β protein expression in striatum (**A**–**D**) and midbrain (**E**–**H**) from A53T-α-syn and WT mice. Each bar represents the mean ± S.E.M. of the percentage of different experimental values obtained in three independent experimental sessions. * *p* < 0.05 compared to WT mice.

**Figure 5 ijms-22-08177-f005:**
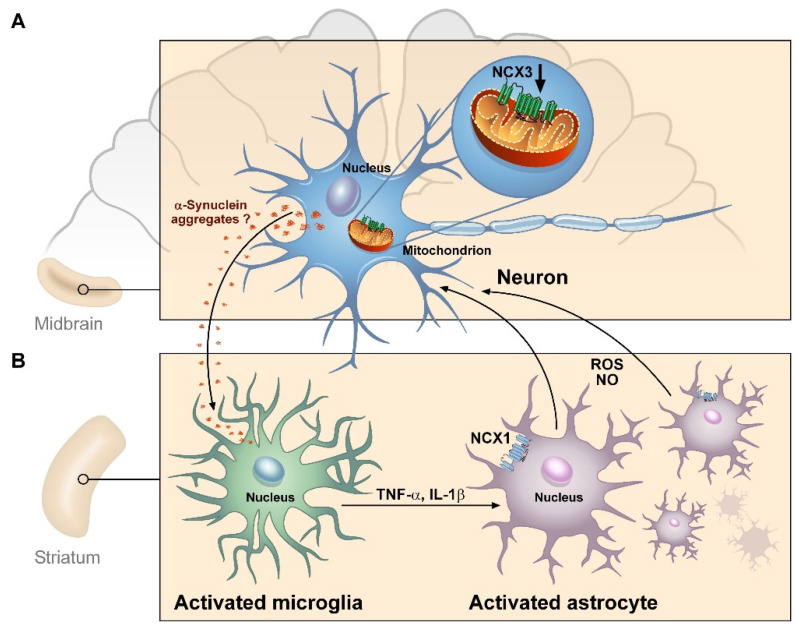
Schematic model illustrating the cellular and molecular events leading to neuroinflammation in A53T-α-syn mice. (**A**) The NCX3 impaired expression in the midbrain leads to mitochondrial dysfunction in dopaminergic neurons with consequent cellular damage and release of toxic factors (probably aggregated α-syn) able to induce microglial cells’ activation in the striatum. (**B**) The microglial activation in the striatum may induce pro-inflammatory cytokines’ release that in turn stimulates NCX1-driven astrocyte proliferation. Finally, the activated astrocytes in the striatum may invade the midbrain, further contributing to dopaminergic neuronal damage.

## Data Availability

The data presented in this study are available on request from the corresponding author. The data are not publicly available due to privacy.

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
