# Peer review of "Ncx3-Induced Mitochondrial Dysfunction in Midbrain Leads to Neuroinflammation in Striatum of A53t-α-Synuclein Transgenic Old Mice"

_ijms, 2021, doi:10.3390/ijms22158177_

Round 1
Reviewer 1 Report
The authors proposed in this work a very interesting study, focusing on the interrelation between α-synuclein pathology, neuroinflammation and intracellular calcium homeostasis where mitochondria play a central role in the development of Parkinson’s neurodegeneration. Interestingly, their results show how the expression and the activity of NCX isoforms, a family of important transporters essential for the cellular ionic homeostasis, are affected by the expression of the mutated variant of alpha-synuclein (A53T) differently in distinct populations of brain cells (neurons and glia).
Using quantitative and functional essays, the authors were able to follow the effect of mitochondrial impairments along the main dopaminergic pathway, from the midbrain area into the striatum in young and aged mice. In such pathological model, they highlight the vicious cycle where dopaminergic neuron firstly are affected by altered NCX3 expression and relative mitochondrial dysfunctions, resulting in activation of microglial cell in the projecting area, the striatum. With time, the activated microglia are able to stimulate neuroinflammation by mean of NCX1-dependent astrocytic proliferation. Such activation would then extend backward to the midbrain area, promoting dopaminergic neurodegeneration. I personally found very elegant the way the authors managed to highlight the role of glia reactivity in parallel with the progression of the mitochondrial pathology, between the presymptomatic stage and the late phase.
It could be interesting to speculate more about the role of the mutation in the mechanism described. Is the A53T variant affecting directly the activity and the expression of NCX3 or it is more likely a secondary effect due to the increasing load of aggregated form of alpha-synuclein?
Although the role of NCX3 in modulating calcium ions in the neurons is well explained in the introduction, the function of NCX1 and the impact of its overexpression on glial cell is not explained but only described as a feature in this A53T model. Is anything known about NCX1 function on glial cells? Has the increased NCX1 expression a detrimental effect on glial functionality or is it just a reflection of glial proliferation? Is it possible to use NCX1 as a therapeutic target to modulate the neuroinflammation in PD?
Minor points:
- In the introduction, next to the citation [16,12] is the term “let’s to hypothesize” mistyped for “let us to hypothesize”?
- In the introduction, the term [Ca2+]m is not explained as well as later in the first paragraph of the results (2.1). Please, add the definition “mitochondrial calcium content” as well.
- In the methods, the last part of the “Western blot analysis” and in the first line for “confocal microscopy and mitochondrial function” the “alpha “symbols are exchanged with another random character. Please, correct them.
- In the last figure, the schematic model, it would be better to exchange the nomenclature “SNpc” with “midbrain”. Unfortunately, the model used in this work (primary cultures) cannot discriminate between SN and VTA dopaminergic neurons. Indeed, it would be very interesting to understand if NCX3 deficit are a distinctive feature of the SNc neurons, but I understand that this is behind the purpose of this work.
Author Response
Comment #1:
“It could be interesting to speculate more about the role of the mutation in the mechanism described. Is the A53T variant affecting directly the activity and the expression of NCX3 or it is more likely a secondary effect due to the increasing load of aggregated form of alpha-synuclein?”
Answer # 1:
We thank the referee for this important suggestion therefore, according to this request we speculated more about the potential role of A53T -syn-mutation in the modulation of NCX3 expression and activity in the discussion’ section: page 7 lines 15-18. Since in striatal neurons and astrocytes obtained from A53T a-syn-mice no changes in NCX3 expression and activity occurs compared to midbrain (See Fig1 and Fig S1), although the increase in a-syn accumulation, it is possible to speculate that the A53T variant might not directly affect NCX3. However, further focused experiments are in progress in order to clarify this aspect, and they will be object of another publication on this topic.
Comment #2:
“Although the role of NCX3 in modulating calcium ions in the neurons is well explained in the introduction, the function of NCX1 and the impact of its overexpression on glial cell is not explained but only described as a feature in this A53T model. Is anything known about NCX1 function on glial cells? Has the increased NCX1 expression a detrimental effect on glial functionality or is it just a reflection of glial proliferation? Is it possible to use NCX1 as a therapeutic target to modulate the neuroinflammation in PD?”
Answer # 2:
We thank the referee for this important suggestion therefore, according to this request we added more details in the discussion with the aim to explain the role of NCX1 in glial cells. See pag 8 lines 54-55, page 9, lines 3-6.
Comment #3:
In the introduction, next to the citation [16,12] is the term “let’s to hypothesize” mistyped for “let us to hypothesize”?
Answer # 3:
We apologize for the mistake that it is now corrected
Comment #4:
In the introduction, the term [Ca2+]m is not explained as well as later in the first paragraph of the results (2.1). Please, add the definition “mitochondrial calcium content” as well.
Answer # 4:
According to Referee suggestion we explained the term [Ca2+]m and added the definition for “mitochondrial calcium content” throughout the manuscript
Comment #5:
In the methods, the last part of the “Western blot analysis” and in the first line for “confocal microscopy and mitochondrial function” the “alpha “symbols are exchanged with another random character. Please, correct them.
Answer # 5:
We apologize for the mistake. We corrected the symbols as required in the revised version of the manuscript
Comment # 6:
In the last figure, the schematic model, it would be better to exchange the nomenclature “SNpc” with “midbrain”. Unfortunately, the model used in this work (primary cultures) cannot discriminate between SN and VTA dopaminergic neurons. Indeed, it would be very interesting to understand if NCX3 deficit are a distinctive feature of the SNc neurons, but I understand that this is behind the purpose of this work.
Answer # 6:
We thank the referee for the suggestion. In the revised version of the manuscript in Fig. 5 we exchanged the nomenclature “SNpc” with “midbrain”. We agree that it would be very interesting to understand if NCX3 deficit are a distinctive feature of the SNc neurons and will try to address this issue in a new work.

Reviewer 2 Report
Di Martino and co-authors investigated the mitochondrial oxidative stress and neuroinflammation in a mouse model of Parkinson’s Disease carring A53T mutation.
The authors described the mitochondrial depolarition in midbrain A53T derived neurons that was not present in A53T derived astrocytes both showing reduced levels of NCX3. Alterations that were not observed in striatal neurons and astrocytes. Midbrain protein extracts showed reduced NCX3 levels only at 4 months of age. A53T mice also exhibited mitochondrial dysfunctions and activation of neuroinflammation. The manuscript may support a role of NCX3 in regulating mitochondrial homeostasis but it needs to be revised. In particular, all throughout the manuscript the authors suggested that the levels of NCX3 directly impact on mitochodrial functions (e.g. line 173-174). The authors should also consider that the aggregation of alpha-synuclein may act on mitochondrial functions leading to NCX3 impairment. Again at line 211-212 it is stated that the results confirm the role of NCX3 in triggering mitochondrial dysfunction but no experiments have been reported in order to evaluate the direct correlation.
- A statement explaining the pathology induced by A53T mutation of asyn should be added in the introduction for non-expert readers.
- Figure 1. It could be useful to add representative pictures of x-rhod1 and fluo-3am.
- Figure 1 caption. Panels B and C reported that primary astrocytes are derived from “mouse pups (1 DIV)”. It is not clear whether the astrocytes have been cultered for 1 day in vitro or 1 DIV is refered to pups 1 day old (the same for Figure S1), in case please correct. Moreover, the statistics should report the astrocytes as well.
- Figure 2. The wb in panel B should be replace with less exposed and clear picture in order to better appreciate the NCX3 reduction.
- Paragraph 2.3. It is not clear why the neuroinflammation has been evaluated only in 16 month-old mice. It would be interesting to understand whether the inflammation and mitochondrial dysfunction appear in earlier or later time point respect to neurodegeneration.
- Figure 4. The wb of IL1beta in the midbrain should be replaced with clearer picture.
- Please verify the symbols in methods (actin and tubulin at the end of wb description, asyn at line 369).
- Please describe the meaning of (Ca2+)i, (Ca2+)m and (Ca2+)c (in the methods) and keep consistent throughout the manuscript.
Author Response
Comment #1:
The manuscript may support a role of NCX3 in regulating mitochondrial homeostasis but it needs to be revised. In particular, all throughout the manuscript the authors suggested that the levels of NCX3 directly impact on mitochodrial functions (e.g. line 173-174). Again, at line 211-212 it is stated that the results confirm the role of NCX3 in triggering mitochondrial dysfunction but no experiments have been reported in order to evaluate the direct correlation.
Answer #1:
As suggested by the Referee we slightly attenuated the tone of the statement indicated in the lines 173-174 and 211-212.
Comment #2:
The authors should also consider that the aggregation of alpha-synuclein may act on mitochondrial functions leading to NCX3 impairment.
Answer #2:
As suggested by the Referee we also considered, in the new version of the manuscript, the hypothesis that the aggregation of alpha-synuclein may act on mitochondrial function leading to NCX3 impairment. See also the answer to Referee#1
Comment #3:
A statement explaining the pathology induced by A53T mutation of asyn should be added in the introduction for non-expert readers.
Answer #3:
We thank the Referee for this important suggestion. Therefore, a statement explaining the pathology induced by A53T mutation of asyn has been added in the introduction for non-expert readers.
Comment #4:
Figure 1. It could be useful to add representative pictures of x-rhod1 and fluo-3am.
Answer #4:
As requested by the Referee Figure 1 has been modified by adding representative pictures of x-rhod1 and fluo-3am.
Comment #5:
Figure 1 caption. Panels B and C reported that primary astrocytes are derived from “mouse pups (1 DIV)”. It is not clear whether the astrocytes have been cultered for 1 day in vitro or 1 DIV is refered to pups 1 day old (the same for Figure S1), in case please correct. Moreover, the statistics should report the astrocytes as well.
Answer #5:
We apologize for the lack of clarity in the caption of figure 1. In the revised version of the manuscript the caption has been modified according to the referee’s suggestion. The same for the figure S1.
Comment #6:
Figure 2. The wb in panel B should be replace with less exposed and clear picture in order to better appreciate the NCX3 reduction.
Answer #6:
We apologize for the picture in Panel B of figure 2. In the revised version of the manuscript the figure in panel B has been exchanged with a clear and less exposed picture n order to better appreciate the NCX3 reduction.
Comment #7:
Paragraph 2.3. It is not clear why the neuroinflammation has been evaluated only in 16 month-old mice. It would be interesting to understand whether the inflammation and mitochondrial dysfunction appear in earlier or later time point respect to neurodegeneration.
Answer #7:
We thank the referee for this important suggestion that has been addressed in the new version of the manuscript both in the results' and in the discussion' sections.
Comment #8:
Figure 4. The wb of IL1beta in the midbrain should be replaced with clearer picture.
Answer #8:
We apologize for the resolution of the images reported in Figure 4. In the revised version of the manuscript, we provided a higher resolution image
Comment #9:
Please verify the symbols in methods (actin and tubulin at the end of wb description, asyn at line 369).
Answer #9:
As suggested by the Referee we verified and corrected the symbols in the methods’ section
Comment #10:
Please describe the meaning of (Ca2+)i, (Ca2+)m and (Ca2+)c (in the methods) and keep consistent throughout the manuscript.
Answer #10:
As suggested by the referee we described the meaning of (Ca2+)i, (Ca2+)m, and (Ca2+)c (in the methods) and kept it consistent throughout the manuscript.

Round 2
Reviewer 1 Report
The authors have satisfactorily responded to all my questions and made the necessary changes to the manuscript.This reviewer considers that the revised version of the manuscript appears to be good and ready for publication in the present form.